# Association of Diet-Related Systemic Inflammation with Periodontitis and Tooth Loss: The Interaction Effect of Diabetes

**DOI:** 10.3390/nu14194118

**Published:** 2022-10-03

**Authors:** Jie Feng, Kun Jin, Xiaomeng Dong, Shi Qiu, Xianglong Han, Yerong Yu, Ding Bai

**Affiliations:** 1State Key Laboratory of Oral Diseases & National Clinical Research Center for Oral Diseases, Department of Orthodontics, West China Hospital of Stomatology, Sichuan University, Chengdu 610041, China; 2Department of Urology, The First Affiliated Hospital of Zhejiang University, Zhejiang University, Hangzhou 310006, China; 3Institute of Urology and National Clinical Research Center for Geriatrics, Department of Urology, West China Hospital, Sichuan University, Chengdu 610041, China; 4Center of Biomedical Big Data, West China Hospital, Sichuan University, Chengdu 610041, China; 5Laboratory of Endocrinology and Metabolism, Department of Endocrinology and Metabolism, West China Hospital, Sichuan University, Chengdu 610041, China

**Keywords:** dietary inflammatory index, periodontitis, tooth loss, diabetes mellitus, National Health and Nutrition Examination Survey

## Abstract

Diet is an important factor that can affect inflammatory processes. Diet-related systemic inflammation is closely linked to periodontitis and tooth loss. However, the role that systemic conditions play in influencing this association remains unclear. A cross-sectional analysis was conducted using the National Health and Nutrition Examination Survey (NHANES) from 2009 to 2014. Diet-related systemic inflammation was assessed by the Dietary Inflammatory Index (DII). Multivariate Cox regression models were used to investigate the association between DII and periodontal results, including total periodontitis, tooth loss, severe tooth loss, and the number of teeth lost. The interaction effects between DII and established covariates were tested. Higher DII scores, corresponding to a higher pro-inflammatory potential of the diet, were associated with an increased risk of periodontitis and tooth loss among the 10,096 eligible participants. There was an interaction between diabetes and DII on total periodontitis (*p* = 0.0136). No significant interaction effect was detected between DII and other established covariates. Participants who consumed an anti-inflammatory diet, and did not have diabetes, experienced the lowest risks of periodontitis and tooth loss. However, in the context of diabetes, the efficacy of such a diet may be weakened or even eliminated. Dietary interventions to manage oral health problems may need to take the individual’s metabolic condition into account.

## 1. Introduction

Periodontitis is a chronic inflammatory disease that affects the tissues surrounding the teeth and is characterized by gradual loss of the supporting tissues of the tooth in combination with halitosis, gingival bleeding, and ultimately tooth loss [1,2]. It affects 11.2% of the world’s population, making it the sixth most common human disease [3]. Population growth trends, improved tooth retention, and changes in risk factors, have increased the socioeconomic burden of periodontitis in recent decades [4]. Interventions to reduce this burden and improve public oral health have attracted considerable attention [5].

The development of dietary strategies may be a cost-effective way of managing periodontal problems [6]. An important feature of periodontitis that has been recognized over the past decade is its relation to systemic inflammation [7,8]. In particular, diet plays an important role in regulating chronic systemic inflammation [9]. Individuals with periodontitis eating a pro-inflammatory diet can not only have exacerbated clinical symptoms of periodontitis but also an increased risk of other systemic diseases such as diabetes mellitus, which can aggravate periodontitis and lead to severe tooth loss [1]. To date, very little information is available on the association of diet-related systemic inflammation with periodontitis and tooth loss, taking into account systemic diseases.

The diet inflammation index (DII) is widely used in different populations to predict inflammation levels and systemic diseases, including cardiovascular diseases, cancer survivors’ mortality, and rheumatoid arthritis [10,11]. Recent large population studies have shown that DII can help predict periodontal outcomes [5]. A higher DII score, corresponding to a pro-inflammatory diet, is significantly associated with greater periodontal probing depth (PPD), clinical attachment (CAL), and additional tooth loss [1,5]. DII has also been demonstrated to mediate the relationship between periodontitis and systemic inflammation [1].

This study aimed to investigate the association of diet-related systemic inflammation with periodontitis and tooth loss, thus demonstrating whether dietary intervention could be incorporated as a protective pattern for the benefit of public oral health.

## 2. Materials and Methods

### 2.1. Study Design and Participants

The study data were obtained from NHANES, which is a stratified, multistage research program for noninstitutionalized civilians in the United States managed by the National Center for Health Statistics [12]. From 2009 to 2014, NHANES implemented a full mouth periodontal examination (FMPE) protocol to collect periodontal measurements for all teeth except the third molar [13]. The examinations were performed in mobile examination centers (MEC) by trained examiners who were calibrated by the survey’s reference examiner. The Centers for Disease Control and Prevention (CDC) and the National Center for Health Statistics (NCHS) Ethics Review Board authorized the NHANES study data. All survey participants gave their informed consent [14]. We complied with the NHANES data user agreement. This study was exempted from the local ethics committee.

For this cohort study, data from complete FMPE cycles (2009–2010, 2011–2012, and 2013–2014) were retrieved and pooled with the following inclusion criteria: participants ≥ 30 years old who underwent FMPE and completed the in-person 24-h interview for total dietary intake. Participants with incomplete data on periodontal assessment, tooth number counts, or dietary recall information were excluded.

The present study was carried out and reported under the guidelines of Strengthening the Reporting of Observational Studies in Epidemiology for Nutritional Epidemiology (STROBE-Nut) [15].

### 2.2. Diet Assessment and DII

DII was used as an exposure variable. DII was developed based on nutritional rationale and derived from the literature-based score to reflect inflammatory potential in an individual’s diet [16]. It was calculated with ‘Inflammatory effect scores’. A total of 45 food parameters and their relation to six inflammatory cytokines, including *C*-reactive protein (CRP), TNF-α IL-1β, IL-4, IL-6, and IL-10, were investigated to evaluate the scores [17,18]. The final DII was calculated by adding all the scores. The DII score’s application and validation were thoroughly documented [19].

The general methodology of DII calculation in this study is described in detail in previous studies [20,21]. Out of the 45 food parameters, 27 were available through NHANES data, including fiber, fat, alcohol intake, carbohydrates, caffeine, cholesterol, omega3, and omega6 PUFA, saturated fatty acids/MUFA/PUFA, protein, magnesium, niacin, zinc, iron, riboflavin, selenium, folic acid, beta carotene, thiamin and vitamins A, B6, B12, C, D, and E. The DII is a continuous measure, with lower (even negative) values indicating a more anti-inflammatory diet, and higher scores indicating a more pro-inflammatory diet. Participants from the total sample were divided into tertiles.

### 2.3. Periodontal Outcome Assessment

FMPE was described at six points per tooth: mesiobuccal, midbuccal, distobuccal, distolingual, midlingual, and mesiolingual (14). In NHANES, periodontitis was classified according to the consensus of the Centers for Disease Control and Prevention (CDC) and Prevention/American Academy of Periodontology for epidemiologic studies [22]. The following classification of periodontitis was used:-Severe periodontitis: PPD ≥ 5 mm in one or more interproximal site(s), and CAL ≥ 6 mm in two or more interproximal sites (not on the same tooth).-Moderate periodontitis: PPD ≥ 5 mm in two or more interproximal sites, with CAL at least 4 mm and <6 mm in two or more interproximal sites (not on the same tooth).-Mild periodontitis: PPD ≥ 4 mm in two or more interproximal sites (or PPD ≥ 5 mm in one site), and CAL at least 3 mm and not ≥4 mm in two or more interproximal sites (not on the same tooth).

Total periodontitis was defined as the accumulation of mild, moderate, and severe periodontitis. The risks of total periodontitis, tooth loss, and severe tooth loss were the primary outcomes. Severe tooth loss was referred to as the number of teeth lost ≥ 10 teeth. The secondary outcome was the number of tooth loss, recorded as the number of sites with dental implant replacements and edentulous sites.

### 2.4. Study Covariates

According to previous research, potential variables that confound the association between DII and periodontal lesions were included in multivariate models that included age, race, sex, body mass index (BMI), marital status, family income ratio to poverty, education level, smoking, and alcohol intake. Study covariates of systemic conditions included diabetes and high blood pressure. The ratio of family income to poverty was developed under the poverty threshold guidelines issued by the US Department of Health and Human Services. A ratio of family income to poverty < 1.00 indicates that the family’s income is below the official poverty level, whereas a ratio of 1.00 or greater shows that the family’s income is prone to exceed the poverty level [23].

Current smokers referred to participants who currently smoked and had smoked at least 100 cigarettes, while former smokers were those who did not smoke now but had smoked more than 100 cigarettes on earlier occasions. Participants who smoked <100 cigarettes were classified as non-smokers [14]. The consumption of alcohol beverages was classified into three groups, defined by the average quantity (drinks/day of drinking) and the average daily drinking volume [11].

### 2.5. Statistical Analyses

The association between DII and periodontal outcomes was established using multivariate adjusted regression. Three progressively adjusted models were developed further based on the initial crude model (model 1, adjusted for age, race, gender, BMI, marital status, the ratio of family income to poverty, and education level; model 2, adjusted for covariates in model 1 plus behavioral variables, including smoking and alcohol intake, and model 3, the fully adjusted model, adjusted for covariates in model 2 along with systemic variables, including diabetes and high blood pressure.)

In regression analysis, interaction effects occur when the influence of one variable is dependent on the value of another variable [23,24]. To test for effect modification, the interaction terms for study covariates and DII were evaluated for each primary outcome with the Cox regression model. Stratified subgroup analyses were performed with multivariate-adjusted regression to detect the influence of interaction effects on the association between DII and periodontal outcomes. Data were analyzed using EmpowerStats (X&Y Solutions, Inc., Boston, MA, USA) and the statistical software packages R (http://www.R-project.org, accessed on 22 September 2021, The R Foundation).

## 3. Results

### 3.1. Characteristics of the Study Population

Of 30,468 participants, 20,372 participants were excluded due to incomplete data about periodontal assessment, the number of teeth lost, and/or dietary intake. The remaining 10,096 participants were included and sorted into tertiles based on their DII value. The prevalence of periodontitis obtained was 39.0% for total periodontitis, 9.4% for severe periodontitis, and 29.6% for mild-moderate periodontitis. Overall, DII scores ranged from −1.21 ± 1.05 in DII-low tertile (median −1.01, range –5.18–0.22), 1.24 ± 0.57 in DII-middle tertile (median 1.25, range 0.22–2.22), to 3.30 ± 0.69 in DII-high tertile (median 3.22, range 2.22–5.42). Participants in the DII-low group were predominantly men (61.75%), while the DII-high group was predominantly women (64.29%, Table 1). Compared to participants in the DII low group, participants in the DII high group were more likely to be non-Hispanic Black, have a family income below the poverty level, have an education limited to high school or lower, and live alone. The DII-high group was more likely to have BMI > 30 kg/m^2^, current smoking status, did not consume alcohol, and have diabetes and/or high blood pressure.

### 3.2. Association of DII and Tooth Loss

Table 2 shows the association between DII and periodontal outcomes. In the unadjusted regression model, a higher DII score was associated with an increased risk of tooth loss, severe tooth loss, and an additional number of tooth loss (pTrend < 0.0001). In adjusted regression models 1–3, the DII-high tertile presented significantly higher risks of tooth loss, severe tooth loss, and additional tooth loss than the DII-low tertile. The differences between the DII-middle and DII-low tertile were marginal. After full adjustment in model 3, the DII-high tertile showed 26% (OR 1.26, 95% CI 1.11–1.43, *p* = 0.0004) and 31% (OR 1.31, 95% CI 1.08–1.60, *p* = 0.0072) higher risks of tooth loss and severe tooth loss than the DII-low tertile. Participants had 0.51 additional missing teeth in the DII-high tertile (95% CI 0.26–0.76, *p* < 0.0001).

### 3.3. Association of DII and Total Periodontitis

A small but significant association was found between DII and total periodontitis based on the unadjusted model (OR 1.05, 95% CI 1.02–1.07, *p* = 0.0001), model 1 (OR 1.04, 95% CI 1.01–1.07, *p* = 0.0076) and model 2 (OR 1.03, 95% CI 1.00–1.06, *p* = 0.0496, Table 2). However, this association was not significant after adjusting for diabetes and high blood pressure in model 3 (OR 1.02, 95% CI 1.00–1.06, *p* = 0.1062). Interaction tests revealed that diabetes influenced the association between DII and total periodontitis after adjustment (*p* = 0.0136, Table 3). No difference was found in diabetes duration (DII-low tertile: median 8 yrs., IQR 3–14 yrs. DII-middle tertile: median 8 yrs., IQR 4–15 yrs. DII-high tertile: median 9 yrs., IQR 4–16 yrs., *p* = 0.139).

### 3.4. Diabetes Modified the Association of DII and Periodontal Outcomes

In subgroup analyses stratified by diabetes status (Table 4), diabetics in the high tertile of DII had an increased risk of tooth loss (OR 1.64, 95% CI 1.08–1.60, *p* = 0.0203) but did not show differences in total periodontitis, severe tooth loss, or the number of teeth lost after adjustment (pTrend > 0.05). For non-diabetic individuals, the association between DII and periodontal outcomes remained significant. Non-diabetic individuals in the DII-high tertile had 16% (OR 1.16, 95% CI 1.02–1.33, *p* = 0.0264), 23% (OR 1.23, 95% CI 1.08–1.40, *p* = 0.0020) and 41% (OR 1.41, 95% CI 1.13–1.77, *p* = 0.0018) higher risks of total periodontitis, tooth loss and severe tooth loss, respectively than their counterparts in the DII-low tertile after adjustment. The individuals without diabetes in the DII-high tertile also had 0.53 additional missing tooth (95% CI 0.27–0.78, *p* < 0.0001).

## 4. Discussion

Our analysis confirms that the level of diet-related inflammation can influence the risks of periodontitis and tooth loss. To the best of our knowledge, this is the first study to explore the joint effect of diabetes and dietary inflammation on periodontitis. It was evident that diabetes had an interactive effect on the association between dietary inflammation and total periodontitis. Diabetes may antagonize the ability of an anti-inflammatory diet to reduce the risks of periodontitis and tooth loss. These findings imply that an anti-inflammatory diet may protect against periodontitis and tooth loss more strongly in the absence of diabetes.

Dietary behavior has long been proposed to influence periodontal health in a clinically relevant way [24]; however, the evidence was inconclusive. Specific nutrients or foods, rather than dietary patterns, could be confounding with the problem of collinearity between some components in meals [25]. Dietary patterns, such as a vegan diet or an anti-inflammatory Mediterranean diet, seemed to influence the salivary microbiota rather than showing a direct relationship to periodontitis [26,27,28]. The findings of previous studies revealed inconsistencies in evaluating the inflammation load of food or the severity of periodontitis [29,30].

DII is a widely used dietary pattern in epidemiological and clinical settings, consistently reflecting the levels of inflammatory markers [1]. The utilization of DII as an instrument to investigate inflammatory diet and periodontal problems has also been verified [1,5]. In a previous study using data from NHANES 2009–2010 and 2011–2012, participants in the upper DII quartile (DII score 2.47 ± 0.6) were shown to have 0.84 (95% CI 0.24–1.45) additional teeth loss [5]. Another study used data from the complete FMPE circles, including NHANES 2009–2010, 2011–2012, and 2013–2014, and found that the association between leukocyte counts and systemic inflammation was mediated by an inflammatory diet [1]. The results of our study support the conception that an anti-inflammatory diet may have protective effects on public oral health. Our results suggest a higher pro-inflammatory potential of the diet was associated with an increased risk of periodontitis and tooth loss, and the subjects who consumed an anti-inflammatory diet and did not have diabetes experienced the lowest risks of periodontitis and tooth loss.

Periodontitis is a well-known complication of diabetic subjects [31]. Hyperglycemia plays an important role in periodontitis, and the level of glycemic control in diabetes dramatically influences the grading of periodontitis [32,33]. A newly published study found that hyperglycemia accelerated inflammation in gingival epithelium through inflammasome activation, and simultaneously induced damage of the gingival epithelial barrier function [34]. Hypoglycemic drugs such as metformin and gliflozins may help reduce the risk of periodontitis in diabetic patients by ameliorating hyperglycemia and attenuation of oxidative stress and inflammation [35]. Diet has been regarded as a novel modifiable factor in regulating systemic inflammatory states [5]. Whether an anti-inflammation diet may help to reduce the risk of periodontitis in patients with diabetes is not known.

In this study we found that diabetic subjects in the higher tertile of DII had an increased risk of tooth loss, but the beneficial efficacy of anti-inflammatory diet observed in non-diabetic subjects was weakened or even eliminated in diabetic subjects. The systemic chronic inflammatory state of diabetes is associated with multiple factors, such as hyperglycemia, lipotoxicosis,, and oxidative stress [36,37]. In addition, more than a third of the patients enrolled in this study had a BMI > 30 kg/cm^2^. Reduced secretion of adiponectin and leptin resistance, associated with adipose tissue expansion, has the potential to exacerbate the chronic inflammatory state in obese diabetic patients [38]. We hypothesize that the combination of many pro-inflammatory factors in diabetic patients is powerful, but the benefits of dietary control alone are difficult to show. This might explain the observed inability of an anti-inflammatory diet to protect against periodontitis in diabetic patients.

The link between diabetes and periodontitis is bidirectional [39]. On the one hand, diabetes is a major cause of periodontitis and subsequent tooth loss [31], and the presence of diabetes may lead to ≥9 additional missing teeth [40]. On the other hand, poor management of periodontitis may affect glycemic control [31]. Specifically, periodontitis is correlated with an increase in glycated hemoglobin (HbA1C), which adversely affects the management of diabetes in such patients [41]. There is insufficient evidence to support periodontitis as a crucial risk factor for diabetes. Periodontitis is deemed an oral complication of uncontrolled diabetes [42]. Other diabetes complications, including heart disease, stroke, diabetic nephropathy, and retinopathy, require long-term observation, during which periodontitis could be managed with regular periodontal care [43]. Management for periodontitis and tooth retention in diabetic patients may follow the principle of diabetic and periodontal treatment rather than dietary regulations [44].

There are limitations to this study. As a cross-sectional study, the causal effect of dietary intervention, systemic inflammation and risk for periodontal diseases need to be validated and extended in prospective studies. Periodontitis and tooth loss were two independent outcomes. Tooth loss could not be attributed to periodontitis in that causes for each missing tooth were not recorded in NHANES. DII was calculated from in-person 24-h recall data, which is inherently biased. In addition, DII reflected a particular diet pattern. Little was known about anti-/pro-inflammatory effects of specific nutrition or foods. The present study was a secondary analysis of a large population-based survey, in which periodontitis prevalence was close to that in a previous report [45]. Although there are many existing and potential confounders that can influence periodontal health, our study confirmed the association of DII with periodontitis and tooth loss. More importantly, greater attention should be paid to the interaction effect of diabetes on this association in the agenda of public health dietary interventions.

## 5. Conclusions

Adherence to a pro-inflammatory diet appears to increase the risks of periodontitis and tooth loss, implying the value of an anti-inflammatory diet as a protective pattern for public oral health. However, in the presence of diabetes, the efficacy of such a diet can be weakened or even eliminated. Dietary interventions to manage oral health problems may need to take the individual’s metabolic condition into account.

## Figures and Tables

**Table 1 nutrients-14-04118-t001:** Characteristics of participants (NHANES 2009–2014, N = 10,096).

	DII Tertiles		
	Low	Middle	High
*n*	3365	3365	3366
DII	−5.18 to 0.22	0.22 to 2.22	2.22 to 5.42
Mean (sd)	−1.21 (1.05)	1.24 (0.57)	3.30 (0.69)
Median (IQR)	−1.01 (−1.86–0.34)	1.25 (0.75–1.74)	3.22 (2.71–3.81)
Age			
Mean (sd)	51.52 (13.99)	51.77 (14.15)	52.64 (14.59)
Median (IQR)	50.00 (40.00–62.00)	51.00 (40.00–62.00)	51.00 (40.00–64.00)
Sex			
Male	2078 (61.75)	1704 (50.64)	1202 (35.71)
Female	1287 (38.25)	1661 (49.36)	2164 (64.29)
Ethnicity			
Mexican American	540 (16.05)	491 (14.59)	414 (12.30)
Other Hispanic	300 (8.92)	329 (9.78)	372 (11.05)
Non-Hispanic White	1546 (45.94)	1475 (43.83)	1419 (42.16)
Non-Hispanic Black	532 (15.81)	700 (20.80)	860 (25.55)
Other Race	447 (13.28)	370 (11.00)	301 (8.94)
Ratio of family income to poverty			
<1.3	748 (24.04)	864 (27.95)	1130 (36.57)
1.3–3.5	1026 (32.98)	1140 (36.88)	1176 (38.06)
>3.5	1337 (42.98)	1087 (35.17)	784 (25.37)
Education level			
Less than high school	589 (17.96)	716 (21.74)	891 (27.25)
High school or GED	590 (17.99)	745 (22.62)	819 (25.05)
Above high school	2101 (64.05)	1832 (55.63)	1560 (47.71)
Marital status			
Married or living with partner	2339 (69.53)	2252 (66.96)	2021 (60.08)
Living alone	1025 (30.47)	1111 (33.04)	1343 (39.92)
BMI (kg/cm^2^)			
<25	962 (28.72)	867 (25.91)	809 (24.19)
25–30	1233 (36.81)	1206 (36.04)	1068 (31.93)
>30	1155 (34.48)	1273 (38.05)	1468 (43.89)
Smoking			
never	1903 (56.55)	1874 (55.71)	1872 (55.66)
former	1059 (31.47)	985 (29.28)	831 (24.71)
current	403 (11.98)	505 (15.01)	660 (19.63)
Alcohol intake			
none	2251 (66.89)	2448 (72.75)	2818 (83.72)
moderate	384 (11.41)	304 (9.03)	201 (5.97)
heavy	730 (21.69)	613 (18.22)	347 (10.31)
Diabetes			
no	2888 (88.35)	2877 (87.79)	2787 (85.20)
yes	381 (11.65)	400 (12.21)	484 (14.80)
High blood pressure			
no	2189 (65.17)	2116 (62.96)	1974 (58.72)
yes	1170 (34.83)	1245 (37.04)	1388 (41.28)

Data are indicated as *n* (%) unless otherwise indicated. DII, dietary inflammation index. GED, general educational development.

**Table 2 nutrients-14-04118-t002:** Risk of periodontal outcomes according to DII tertile groups (N = 10,096).

	Non-Adjusted	Model 1	Model 2	Model 3
Total periodontitis				
DII-low	Reference	Reference	Reference	Reference
DII-middle	1.08 (0.98, 1.18) 0.1305	1.10 (0.98, 1.23) 0.1096	1.08 (0.96, 1.22) 0.1756	1.08 (0.96, 1.21) 0.2194
DII-high	1.21 (1.10, 1.33) 0.0001	1.18 (1.04, 1.33) 0.0076	1.13 (1.00, 1.28) 0.0496	1.11 (0.98, 1.25) 0.1062
*pTrend*	0.0001	0.0073	0.0473	0.101
Tooth loss				
DII-low	Reference	Reference	Reference	Reference
DII-middle	1.16 (1.05, 1.29) 0.0039	1.12 (1.00, 1.26) 0.0472	1.12 (1.00, 1.25) 0.0595	1.12 (1.00, 1.26) 0.0571
DII-high	1.52 (1.37, 1.69) <0.0001	1.28 (1.13, 1.45) <0.0001	1.25 (1.10, 1.42) 0.0004	1.26 (1.11, 1.43) 0.0004
*pTrend*	<0.0001	<0.0001	0.0004	0.0004
Severe tooth loss				
DII-low	Reference	Reference	Reference	Reference
DII-middle	1.08 (0.91, 1.28) 0.3980	0.99 (0.81, 1.21) 0.9075	0.97 (0.79, 1.18) 0.7399	0.97 (0.78, 1.19) 0.7420
DII-high	1.61 (1.37, 1.89) <0.0001	1.35 (1.12, 1.64) 0.0020	1.29 (1.06, 1.56) 0.0110	1.31 (1.08, 1.60) 0.0072
*pTrend*	<0.0001	0.0012	0.0075	0.0046
Numbers of tooth loss				
DII-low	Reference	Reference	Reference	Reference
DII-middle	0.28 (0.04, 0.52) 0.0245	0.08 (−0.16, 0.32) 0.5057	0.05 (−0.19, 0.28) 0.6972	0.05 (−0.19, 0.29) 0.6952
DII-high	1.16 (0.92, 1.41) <0.0001	0.58 (0.33, 0.83) <0.0001	0.48 (0.23, 0.73) 0.0002	0.51 (0.26, 0.76) <0.0001
*pTrend*	<0.0001	<0.0001	0.0003	0.0001

Data are indicated as OR (95% CI) with *p* value unless otherwise indicated. Model 1 adjusted for age, ethnicity, sex, BMI, marital status, ratio of family income to poverty line, education level. Model 2 adjusted for the same variables as model 1, as well as smoking and alcohol intake. Model 3 adjusted for the same variables as model 2, as well as diabetes and high blood pressure. DII, dietary inflammation index.

**Table 3 nutrients-14-04118-t003:** Stratified logistic regression analysis to identify variables that have interaction effects between DII and risk of periodontal outcomes.

Variable	*n*		Periodontal Outcomes	*p* for Interaction	
	Crude Model	Adjusted Model		Crude Model	Adjusted Model
Age					
<50 years	4763	4763	Total periodontitis	0.7407	0.6908
≥50 years	5333	5333	Tooth loss	0.9807	0.8334
			Severe tooth loss	0.8796	0.7994
Sex					
Male	4984	4984	Total periodontitis	0.7601	0.6468
Female	5112	5112	Tooth loss	0.9616	0.8462
			Severe tooth loss	0.7663	0.9798
BMI (kg/cm^2^)					
<25	2638	2638	Total periodontitis	0.2703	0.4551
25–30	3507	3507	Tooth loss	0.482	0.7228
>30	3896	3896	Severe tooth loss	0.8933	0.9184
Smoking					
never	5649	5649	Total periodontitis	0.0019	0.6506
former	2875	2875	Tooth loss	0.4724	0.1088
current	1568	1568	Severe tooth loss	0.0371	0.6329
Alcohol intake					
none	7517	7517	Total periodontitis	0.2719	0.6854
moderate	889	889	Tooth loss	0.374	0.4698
heavy	1690	1690	Severe tooth loss	0.9007	0.8551
Diabetes					
no	8552	8552	Total periodontitis	0.0149	0.0144
yes	1265	1265	Tooth loss	0.0866	0.1292
			Severe tooth loss	0.2512	0.4587
High blood pressure					
no	6279	6279	Total periodontitis	0.3984	0.6381
yes	3803	3803	Tooth loss	0.5705	0.8891
			Severe tooth loss	0.578	0.6852

No interaction effects were found for socioeconomic covariates including ethnicity, marital status, ratio of family income to poverty line or education level (data not shown). The adjusted model adjusted for age, ethnicity, sex, BMI, marital status, ratio of family income to poverty line, education level, smoking, alcohol intake, diabetes and high blood pressure. In each stratification, the model is not adjusted for the stratification variable.

**Table 4 nutrients-14-04118-t004:** Multi-regression analysis of DII on periodontal outcomes stratified according to diabetes.

	Diabetic Patients		Non-Diabetic Participants	
	*n* = 1265		*n* = 8552	
	Non-Adjusted	Adjusted	Non-Adjusted	Adjusted
Total periodontitis				
DII-low	Reference	Reference	Reference	Reference
DII-middle	1.15 (0.85, 1.57) 0.3611	1.22 (0.85, 1.74) 0.2725	1.06 (0.96, 1.17) 0.2756	1.06 (0.94, 1.20) 0.3390
DII-high	0.88 (0.66, 1.17) 0.3847	0.80 (0.57, 1.13) 0.2107	1.23 (1.11, 1.37) <0.0001	1.16 (1.02, 1.33) 0.0252
*pTrend*	0.3535	0.2077	0.0001	0.0264
Tooth loss				
DII-low	Reference	Reference	Reference	Reference
DII-middle	0.94 (0.67, 1.33) 0.7417	0.93 (0.63, 1.37) 0.7122	1.19 (1.06, 1.32) 0.0020	1.14 (1.01, 1.29) 0.0342
DII-high	1.77 (1.24, 2.53) 0.0018	1.64 (1.08, 2.49) 0.0203	1.47 (1.31, 1.65) <0.0001	1.23 (1.08, 1.40) 0.0024
*pTrend*	0.0020	0.0274	<0.0001	0.0020
Severe tooth loss				
DII-low	Reference	Reference	Reference	Reference
DII-middle	1.22 (0.83, 1.79) 0.3073	0.92 (0.60, 1.42) 0.7106	1.04 (0.85, 1.28) 0.6834	0.98 (0.77, 1.25) 0.8818
DII-high	1.41 (0.99, 2.01) 0.0557	1.01 (0.66, 1.53) 0.9787	1.67 (1.38, 2.02) <0.0001	1.41 (1.13, 1.77) 0.0029
*pTrend*	0.0553	0.9556	<0.0001	0.0018
Numbers of tooth loss				
DII-low	Reference	Reference	Reference	Reference
DII-middle	0.41 (−0.49, 1.31) 0.3764	−0.18 (−1.09, 0.74) 0.7025	0.25 (0.00, 0.49) 0.0490	0.08 (−0.16, 0.32) 0.5284
DII-high	1.40 (0.54, 2.26) 0.0015	0.45 (−0.46, 1.37) 0.3320	1.08 (0.83, 1.33) <0.0001	0.53 (0.27, 0.78) <0.0001
*pTrend*	0.0014	0.3334	<0.0001	<0.0001

Data are indicated as OR (95% CI) with *p* value unless otherwise indicated. The adjusted model adjusted for age, ethnicity, sex, BMI, marital status, ratio of family income to poverty line, education level, smoking, alcohol intake and high blood pressure. DII, dietary inflammation index.

## Data Availability

Data are available at the NHANES website.

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
