# Peer review of "Association of Diet-Related Systemic Inflammation with Periodontitis and Tooth Loss: The Interaction Effect of Diabetes"

_nutrients, 2022, doi:10.3390/nu14194118_

Round 1

Reviewer 1 Report

This work covers a very relevant hot topic in diet & dentistry sciences and presents clearly an association between inflammatory index, diabetes and periodontitis. Therefore further comments intend to help improving its contribute to scientific evidence.  

Line 17 Not sure if you can say it is ambiguous, you cite a recent published work showing this association (reference 1).  

Line 74 NHANES had been approved by an ethic committee. However, because you use results from their open database you just have to refer and comply with data user agreement. It is not because data is anonymous that ethical approval can be dismissed.  

Line 78 Considering that you use NHANES database, after 2002 a second 24h dietary recall add been added. Do you have access to this information? That is in fact the highest limitation when you refer to this method in line 270. Three dietary 24h recalls are needed to proper assess energy, fiber and protein intake at least and you just refer one.  

Line 238 The (1) is a reference or typing error? 

Line 143 - Reference 24 seems erroneous used, not related with the statistical association you are suggesting. 

Line 216 In spite it is the first analysis and manuscript about this subject, it is not a clinical trial rather an association study from population-based results and that could improve methods (like dietary assessment). Therefore, I would suggest some moderation in this statement. You can say it is the first time this association is explored.  

Author Response

Reviewer 1

This work covers a very relevant hot topic in diet & dentistry sciences and presents clearly an association between inflammatory index, diabetes and periodontitis. Therefore further comments intend to help improving its contribute to scientific evidence.  

Line 17 Not sure if you can say it is ambiguous, you cite a recent published work showing this association (reference 1).  

RESPONSE: The authors thank the reviewer for the feedback. We deem that the study in “reference 1” have demonstrated the mediation effect of inflammatory diet on the association of leukocyte counts and systemic inflammation with PD, but not how systemic inflammation impacted the association of inflammatory diet with PD. We intended to throw light upon how systemic conditions (diabetes) influence such association, which is currently not well understood. In order to avoid the “ambiguous” expression, we replaced the word by “unclear” to the Abstract on page 1, line 17.

Revised text

However, the role that systemic conditions play in influencing this association remains unclear.

Line 74 NHANES had been approved by an ethic committee. However, because you use results from their open database you just have to refer and comply with data user agreement. It is not because data is anonymous that ethical approval can be dismissed.  

RESPONSE: The authors thank the reviewer for the helpful suggestion. We have made amendment to the Materials and Methods on page 2, line 73.

Revised text

We complied with NHANES data user agreement. This study was exempted from the local ethics committee.

Line 78 Considering that you use NHANES database, after 2002 a second 24h dietary recall add been added. Do you have access to this information? That is in fact the highest limitation when you refer to this method in line 270. Three dietary 24h recalls are needed to proper assess energy, fiber and protein intake at least and you just refer one.  

Revised text

  • For this cohort study, data from complete FMPE cycles (2009–2010, 2011–2012, and 2013–2014) were retrieved and pooled with the following inclusion criteria: participants ³ 30 years old, underwent FMPE and completed the in-person 24-hour interview for total dietary intake.
  • DII was calculated from the in-person 24-hour recall data, which is inherently biased.

Line 238 The (1) is a reference or typing error? 

RESPONSE: The authors thank the reviewer for the feedback. It is a typing error, and we must apologize for this mistake. We have deleted it and carefully checked the whole manuscript.

Line 143 - Reference 24 seems erroneous used, not related with the statistical association you are suggesting. 

RESPONSE: The authors thank the reviewer for the feedback. We have deleted this reference.

Line 216 In spite it is the first analysis and manuscript about this subject, it is not a clinical trial rather an association study from population-based results and that could improve methods (like dietary assessment). Therefore, I would suggest some moderation in this statement. You can say it is the first time this association is explored.  

RESPONSE: The authors thank the reviewer for the insightful suggestion. We agree with the reviewer that this study is not a clinical trial to prove the association of dietary assessment and periodontal outcomes. Rather it is an observational study to explore such association. We have made amendment to the Discussion on page 8, line 217.

Revised text

To the best of our knowledge, this is the first study to explore the joint effect of diabetes and dietary inflammation on periodontitis.

Reviewer 2 Report

I read with great interest the paper “Association of diet-related systemic inflammation with periodontitis and tooth loss: the interaction effect of diabetes " by Feng et al.

Paper design is fine. The article is logically divided into sections and subsections. English is fine, only minor spell check needed. Statistical analysis is fine.

Comments:

1.      More than a third of the patients enrolled have a BMI>30. it is fundamental to underline the role of the adipose tissue as an endocrine tissue, able of producing several hormones such as adiponectine and leptin, which plays a major role in inflammation. In fact, Leptin is produced by the adipose tissue in proportion to its mass and, in larger quantities, it acts as a proinflammatory hormone (doi: 10.37349/emed.2020.00019). Please discuss it.

2.      Diabetes duration is not reported. In fact, the complication onset in diabetes are associated to disease duration. Please report such data. If not available, it should be reported in the limit section.

3.      Discussion, line 254-259: This part should be better clarified. In fact, hyperglycaemia plays a crucial role in accelerated inflammation in the gingival epithelium through inflammasomes activation, which is potentially affiliated with a decline in the gingival epithelial barrier function in diabetes. Moreover, nothing is said about pharmacological treatment, which can ameliorate hyperglycaemia, oxidative stress, inflammation, and body weight. In fact, drugs such as metformin and gliflozins also shows this effects (doi: 10.1111/jre.12863; doi: 10.3390/biom11121834).

Author Response

Dear Editor,

On behalf of my co-authors, I would like to thank the reviewers and editors for their consideration of our work and their constructive feedback. As requested, we have revised our manuscript and herein enclose the revised version (with changes tracked), together with point-by-point responses to the comments. We greatly appreciate the opportunity to revise our work and hope that our manuscript is now acceptable for publication in Nutrients. We look forward to your response.

The answers to each of the reviewers’ comments are listed in the following format.

RESPONSE: The authors’ response to each comment is listed verbatim.

Revised text: The corresponding lines in the manuscript containing the revised text are noted. The revised text in response to the reviewer’s comment is quoted below the response.

Response to reviewers

Review2

I read with great interest the paper “Association of diet-related systemic inflammation with periodontitis and tooth loss: the interaction effect of diabetes " by Feng et al.

Paper design is fine. The article is logically divided into sections and subsections. English is fine, only minor spell check needed. Statistical analysis is fine. 

Comments:

  • More than a third of the patients enrolled have a BMI>30. it is fundamental to underline the role of the adipose tissue as an endocrine tissue, able of producing several hormones such as adiponectine and leptin, which plays a major role in inflammation. In fact, Leptin is produced by the adipose tissue in proportion to its mass and, in larger quantities, it acts as a proinflammatory hormone (doi: 10.37349/emed.2020.00019). Please discuss it.

RESPONSE: The authors thank the reviewer for the insightful suggestion. We have made major revision to the manuscript (the Discussion on page 8, line 257).

Revised text

The systemic chronic inflammatory state of diabetes is associated with multiple factors, such as hyperglycemia, lipotoxicosis,, and oxidative stress [36, 37]. In addition, more than a third of the patients enrolled in this study have a BMI>30 kg/cm2. Reduced secretion of adiponectin and leptin resistance, associated with adipose tissue expansion, has the potential to exacerbate the chronic inflammatory state in obese diabetic patients [38]. We hypothesize that the combination of many pro-inflammatory factors in diabetic patients are powerful that the benefits of dietary control alone are difficult to show.

  1. Woelber JP, Bremer K, Vach K, et al. An oral health optimized diet can reduce gingival and periodontal inflammation in humans - a randomized controlled pilot study. BMC Oral Health. 2017, 17, 28.
  2. Beverly JK, Budoff MJ. Atherosclerosis: Pathophysiology of insulin resistance, hyperglycemia, hyperlipidemia, and inflammation. Journal of Diabetes. 2020, 12, 102-104.
  3. Acierno C, Caturano A, Pafundi PC, Nevola R, Adinolfi LE, Sasso FC. Nonalcoholic fatty liver disease and type 2 diabetes: pathophysiological mechanisms shared between the two faces of the same coin. Explor Med. 2020, 1, 287-306.
  • Diabetes duration is not reported. In fact, the complication onset in diabetes are associated to disease duration. Please report such data. If not available, it should be reported in the limit section.

RESPONSE: The authors thank the reviewer for the helpful feedback. We have reprocessed the data to investigate diabetes duration among DII tertiles. And no difference was found among DII-low, DII-middle and DII-high groups. The findings have been added to the Results on page 5, line 186.

Revised text

No difference was found in diabetes duration among DII titles (DII-low tertile: median 8 yrs., IQR 3-14 yrs. DII-middle tertile: median 8 yrs., IQR 4-15 yrs. DII-high tertile: median 9 yrs., IQR 4-16 yrs., p=0.139).

  • Discussion, line 254-259: This part should be better clarified. In fact, hyperglycaemia plays a crucial role in accelerated inflammation in the gingival epithelium through inflammasomes activation, which is potentially affiliated with a decline in the gingival epithelial barrier function in diabetes. Moreover, nothing is said about pharmacological treatment, which can ameliorate hyperglycaemia, oxidative stress, inflammation, and body weight. In fact, drugs such as metformin and gliflozins also shows this effects (doi: 10.1111/jre.12863; doi: 10.3390/biom11121834).

RESPONSE: The authors thank the reviewer for the insightful suggestion. We have made major revision to the manuscript (the Discussion on page 9, line 245).

Revised text

Periodontitis is a well-known complication of diabetic subjects [31]. Hyperglycemia plays an important role in periodontitis, and the level of glycemic control in diabetes dramatically influences the grading of periodontitis [32, 33]. A newly published study found that hyperglycemia accelerated inflammaging in gingival epithelium through inflammasome activation and simultaneously induced the damage of gingival epithelial barrier function [34]. Hypoglycemic drugs such as metformin and gliflozins may help reduce the risk of periodontitis in diabetic patients by ameliorating hyperglycemia and attenuation of oxidative stress and inflammation [35].

  1. Genco RJ, Borgnakke WS. Diabetes as a potential risk for periodontitis: association studies. Periodontol 2000. 2020, 83, 40-45.
  2. Tonetti MS, Greenwell H, Kornman KS. Staging and grading of periodontitis: Framework and proposal of a new classification and case definition. J Periodontol. 2018, 89, S159–72.
  3. Jepsen S, Suvan J, Deschner J. The association of periodontal diseases with metabolic syndrome and obesity. Genco R, editor. Periodontol 2000. 2020, 83, 125–53.
  4. Zhang P, Lu B, Zhu R, Yang D, Liu W, Wang Q, et al. Hyperglycemia accelerates inflammaging in the gingival epithelium through inflammasomes activation. J Periodont Res. 2021, 56, 667–78.
  5. Salvatore T, Galiero R, Caturano A, Vetrano E, Rinaldi L, Coviello F, et al. Effects of Metformin in Heart Failure: From Pathophysiological Rationale to Clinical Evidence. Biomolecules. 2021, 11, 1834.

Round 2

Reviewer 2 Report

The paper has much improved. The implemented results with diabetes duration provides a deeper insight into diabetic complications and give more prominence to the paper. All the issue I raised have been rightfully addressed. The paper can now be further processed for publication.